# Composite core set construction and diversity analysis of Iranian walnut germplasm using molecular markers and phenotypic traits

Razieh Mahmoodi[1,2], Mohammad Reza Dadpour[1]*, Darab Hassani[2]*, Mehrshad Zeinalabedini[3], Elisa Vendramin[4], Charles A. Leslie[5]

1 Department of Horticulture Sciences, Faculty of Agriculture, University of Tabriz, Tabriz, Iran, 2 Temperate Fruits Research Center, Horticultural Science Research Institute, Agricultural Research, Education and Extension Organization (AREEO), Karaj, Iran, 3 Systems Biology Department, Agricultural Biotechnology Research Institute of Iran, Agricultural Research, Education and Extension Organization (AREEO), Karaj, Iran, 4 Centro di Ricerca per l'Olivicoltura, Frutticoltura e Agrumicoltura, Consiglio per la Ricerca in Agricoltura e l'Analisi dell'Economia Agraria, Roma, Italy, 5 Department of Plant Sciences, University of California, Davis, CA, United States of America

* d.hassani@areeo.ac.ir (DH); dadpour.mr@gmail.com (MRD)

**Data Availability Statement:** All relevant data are within the paper and its Supporting Information files.

## Abstract

Iran is a center of origin and diversity for walnuts (*Juglans regia* L.) with very good potential for breeding purposes. The rich germplasm available, creates an opportunity for study and selection of the diverse walnut genotypes. In this study, the population structure of 104 Persian walnut accessions was assessed using AFLP markers in combination with phenotypic variability of 17 and 18 qualitative and quantitative traits respetively. The primers E-TG/M-CAG, with high values of number of polymorphic bands, polymorphic information content, marker index and Shannon's diversity index, were the most effective in detecting genetic variation within the walnut germplasm. Multivariate analysis of variance indicated 93.98% of the genetic variability was between individuals, while 6.32% of variation was among populations. A relatively new technique, an advanced maximization strategy with a heuristic approach, was deployed to develop the core collection. Initially, three independent core collections (CC1–CC3) were created using phenotypic data and molecular markers. The three core collections (CC1–CC3) were then merged to generate a composite core collection (CC4). The mean difference percentage, variance difference percentage, variable rate of coefficient of variance percentage, coincidence rate of range percentage, Shannon's diversity index, and Nei's gene diversity were employed for comparative analysis. The CC4 with 46 accessions represented the complete range of phenotypic and genetic variability. This study is the first report describing development of a core collection in walnut using molecular marker data in combination with phenotypic values. The construction of core collection could facilitate the work for identification of genetic determinants of trait variability and aid effective utilization of diversity caused by outcrossing, in walnut breeding programs.

**Funding:** The author(s) received no specific funding for this work.

**Competing interests:** The authors have declared that no competing interests exist.

## Introduction

Persian walnut (*Juglans regia* L.), is the most important species of the *Juglandaceae* family for its valuable nuts. Iranian plateau has been considered as a center of origin and domestication of this species, where it still exhibits a great diversity. In overall, sexual propagation of the walnut might be the main reason for high genetically variation which still exists among natural populations. Thoroughly, economic and nutritional value of the Persian walnut has been lead to world-wide distribution of the species especially in temperate regions. As it is known, the walnut is a monoicous species. The existence of protandry which usually cause the outcrossing, increase the variability and affects the population structure. This phenomenon together with the sexual propagation, created a huge segregated walnut population in Iran.

During the recent decades, there has been remarkable progress in evaluation of walnut germplasm and establishment of a collection in Iran. Basically, germplasm collections are considered as a valuable sources of conserving genetic diversity and providing plant material for breeders, but in collections with very high number of accessions, it can be more expensive to identify the appropriate stock. On the other hands, management of a large number of individuals could be effectively difficult [1]. To overcome this issue, Brown [1] proposed the concept of core collection. Several methods have been exploited to develop core collections. Initially, most researchers performed random sampling [2]. More recently, progresses in molecular biology have facilitated establishment of core collections using molecular markers, either alone [3] or with phenotypic traits [4–7]. The maximization strategy has been developed using PowerCore v. 1.0 software upon which, selection of accessions with the highest diversity could be possible [8]. Up to now, core collections have been established for a number of fruit tree species [4, 9, 10].

Regarding to the Persian walnut, the first core collection was established using phenological traits [11]. Since phenotypic traits are substantially influenced by environmental factors, this research has been used as complementary to create a more robust core collection. The main reason for pooling the phenotypic data and molecular procedures together is that, the molecular markers can only reflect variation at DNA level that is not necessarily expressed in the phenotype [12].

During recent decades, many studies focused on the genetic diversity and population structure of Persian walnut based on SSR [13], RAPD [14], AFLP [15]and SNP [16].

This study tried to describe the genetic diversity and population structure of a collection of 104 walnut accessions in order to assume variation and to identify a robust core collection. To the best of our knowledge, this is the first report for utilizing molecular variation and phenotypic data in walnut.

## Materials and methods

The main collection used in this study has been established at 35.754888˚ N and 50.952986˚ E in Karaj in 2006. For the establishment of the main collection, in the first step, pre-selections were done among more than 10000 walnut genotypes from the walnut producing areas including Karaj, Qazvin, Tabriz, Urmia, Kerman, Tuyserkan, and Shahroud from 2003 to 2004. Subsequently, the 104 walnut genotypes selected based on phenotypic traits, were propagated by grafting and were planted together with nine foreign walnut cultivars: Chandler, Pedro, Hartley, Serr, Howard, Ronde de Montignac, Alsozentivani 117, Frjean and Roxana, in the main walnut collection. The 104 walnut genotypes were belonging to seven autochthonous origin (Alborz, Kerman, Qazvin, Shahroud, Tabriz, Touyserkan and Urmia), and two foreign groups (USA and Europe). Details, including accession ID, name, and origin, are described in Mahmoodi et al [11].

## Measurement of phenotypic data

The accessions were evaluated for 17 qualitative traits (nut size, nut shape in longitudinal section through suture, nut shape in longitudinal section perpendicular to suture, nut shape in cross section, nut shape of base perpendicular to suture, shape of apex perpendicular to suture, prominence of apical tip, position of pad on suture, prominence of pad on suture, width of pad on suture, depth of grove along pad on suture, structure of surface of shell, adherence of two halves of shell, thickness of dividing membranes, ease of removal, intensity of ground color and kernel size), 18 quantitative traits (bud break; start, end and duration of pollen shedding and pistillate flowers receptivity; nut length, width, thickness and roundness; nut and kernel weight; kernel percentage; shell and membrane thickness; and number of nuts to scaffold and tranck cross area) [11] and AFLP markers. The measurements of nut and kernel traits were based on 20 nut samples. The descriptor for qualitative traits, were reported in S1 Table. The Shannon diversity index (H), a parameter commonly used to characterize diversity in populations, was calculated based on Shannon [17].

## Genomic DNA extraction and AFLP analysis

For determining the genetic variability among the walnut genotypes, 13 AFLP primers were evaluated. Genomic DNA was extracted from 200 mg of leaf tissue of each accession using the CTAB method of Doyle and Doyle with minor modifications [18]. Based on the results, concentrations of DNA extracts were standardized for AFLP analysis. The AFLP was performed using the method of Vos et al. [19] with modifications, using enzyme combination EcoRI/MseI. The AFLP primer combinations (MseI: 3 selective nucleotides, EcoRI:2 selective nucleotides) were labeled with infrared dyes IRD-700 and IRD-800 at the 5´ end, accompanied by three and two selective nucleotides at the 3´end. Briefly, 5 μl of extracted DNA at a concentration of approximately 50 ng/ μl genomic DNA was digested with EcoRI/MseI (1 U) and incubated at 37˚C for 3 h. The fragments were ligated with T4 DNA ligase to EcoRI and MseI adapters at 37˚C for 3 h followed by 4˚C overnight. Ligated DNA was diluted 1:5 with water and used for pre-amplification. The pre-amplification reactions were performed using non-selective primers (E000 and M000) in a 25 μl reaction volume (containing: 3.75 μl of (1:3) diluted ligation product, 1 unit of Taq polymerase, 1X Taq polymerase buffer, 0.4 μM of each of the two primers, 150 μM of each of dATP, dCTP, dGTP and dTTP, and 2 mM MgCl2). This amplification was performed using the following cycling parameters: 25 cycles, each consisting of 1 min at 94˚C, 1 min at 60˚C, 2 min at 72˚C and final extension was done at 72˚C for 7 min. The preamplification products were diluted in the ratio 1:9 by sterile distilled water.

Selective amplification was performed using reaction amplification performed in a 25 μl reaction mixture volume containing: 3.75 μl of diluted pre-amplification product, 1x Taq polymerase buffer, 2 mM MgCl2, 1 Unit of Taq polymerase, 150 μM of dNTPs, and 0.4 μM of each of the two primers with two or three additional nucleotides at the 3´end.

PCR program was continued by 10 cycle of 94˚C for 3 min for pre per denaturation and followed by 30 s at 95˚C, 30 s at 63˚C as touchdown with 1˚C lowering for each cycle, 2 min at 72˚C. The amplified products were run on a 6.5% polyacrylamide gel using DNA analyzer (LI-COR 4300, USA). The Sequences of all adaptors and primers are listed in S2 Table.

## Data analysis

Amplified AFLP products were assembled into a binary matrix by scoring each fragment manually as presence (1) or absence (0) of a band across all 104 walnut cultivars for each primer combination. The variability parameters were assessed using POPGENE version 1.32 [20]. To determine which of the AFLP primer combinations has most effectively differentiated the

genotypes, polymorphism information content (PIC), marker index (MI) and resolving power (RP) were calculated [21].

To investigate genetic differentiation among the walnut populations, analysis of molecular variance (AMOVA) was performed using Arlequin 3.11 software [22]. The analysis could be performed using intra-population and inter-population methods. In the first option statistical information would be extracted independently from each population, whereas in the second method, samples would be compared to each other.

To determine phylogenetic relationships among accessions, clustering analysis based on the repeated bisection (RB) method was performed using gCLUTO software (version 1.0, University of Minnesota, Twin Cities, MI, USA). This is a graphical application for clustering low- and high-dimensional datasets and analyzing the characteristics of the clusters. Principal coordinate analyses were also performed using GenAlEx ver 6.502 [23].

STRUCTURE software ver. 2.3.4 was used to analyze the population structure of the full germplasm collection. The number of clusters was selected after 10 independent runs of a burn-in period of 100,000 iterations and 100,000 MCMC repetitions for each value of K (k = 1–10). The optimum value of K was obtained by calculating the ΔK and highest LnP(D) value to determine the most likely number of groups [24]. The results from STRUCTURE were processed with the online software STRUCTURE HARVESTER v.0.6.1 to obtain the most acceptable K value.

## Development of core collection

A core collection was developed using PowerOWERCore program [8], genotypic data and 17 qualitative traits (S1 Table). Four methods were used to determine core collection options; 1) genetic analysis of the entire collection, 2) phenotypic analysis based on qualitative traits of the entire collection, 3) phenotypic analysis based on quantitative traits of the entire collection and, 4) a combination of phenotypic and molecular variability by merging core collections. Categorical variables (genetic and phenotypic) were applied based on distinct characters. Continuous variables, i.e. quantitative traits, were classified into different categories by the software, based on Sturges' rule [25].

## Evaluation of the core collections

To evaluate the ability of each proposed core set to represent the full collection, the Mean Difference, Coincidence Rate, Variance Difference, Variable rate and Coverage (%) were calculated [16].

In addition, Shannon's diversity index (I) and Nei's gene diversity (H) values were calculated using POPGENE version 1.32 [20].

For a core collection to be considered representative of its primary collection, MD% should be less than 20% and CR% more than 80%. Lower VD values and higher VR values indicate a more effective core collection [8]. The core coverage CR% should exceed 80% of the full collection [26].

Principal coordinate analysis was used to assess segregation patterns in the full and core collections.

## Results

Among the 13 evaluated primers, five primer combinations showed polymorphism with a total of 499 total and 197 polymorphic fragments (Table 1). The primer pair E-TG/M-CAG was the most efficient in discriminating the individuals with a polymorphism rate of 52.08%. The least discriminatory primer was the pair of E- CT/M-GAG with a polymorphism rate of

**Table 1. Variability parameters for five AFLP primer combinations.**

| Primer combinations | TNB | NPB | %PPB | RP | PIC | MI | I |
|---|---|---|---|---|---|---|---|
| E-TG(IR800)/M-GAG | 98 | 43 | 43.43 | 17.31 | 0.154 | 2.876 | 0.229 |
| E-TG(IR800)/M-CAG | 96 | 50 | 52.08 | 14.92 | 0.169 | 4.375 | 0.254 |
| E-AT(IR700)/M-GAG | 98 | 39 | 39.00 | 12.13 | 0.136 | 2.069 | 0.205 |
| E-CT(IR800)/M-GAG | 99 | 29 | 29.29 | 12.13 | 0.106 | 0.90 | 0.157 |
| E-TG(IR800)/M-CAT | 108 | 36 | 33.33 | 16.06 | 0.129 | 1.536 | 0.188 |
| Mean | | 39.40 | 39.43 | 14.508 | 0.139 | 2.351 | 0.207 |

**TNB**: Total Number of Bands, **NPB**: Number of Polymorphic Bands, **PPB**: Percentage of Polymorphic Bands, **PIC**: Polymorphic Information Content, **MI**: Marker Index, **I**: Shannon's Information Index.

29.29%. For dominant markers such as AFLP, estimation of marker index, together with PIC value, has been used to assess the degree of informativeness of markers [21]. In this study, the range of PIC was varied from 0.106 to 0.169 with a general mean of 0.139. The marker index value (MI) for each primer pair was computed too. The mean value for MI in this study was 2.35 with a range from 0.90 (E- CT/M-GAG) to 4.37 (E-TG/M- CAG). In addition, the Shannon's information index (I) was in concordance with PIC and MI. The highest and lowest Shannon index were belong to primer E-TG/M- CAG (I = 0.254) and CT/M-GAG (I = 0.157) respectively (Table 1). The Resolving power (RP) varied from 12.13 to 17.31. In summary, the primer pair E-TG/M-CAG was found to be the most effective in detecting genetic variation among walnut germplasm. Genetic diversity parameters for 9 populations are shown in Table 2. According to the results, the highest number of polymorphic loci (NP) was obtained in the Qazvin population (37.05% polymorphic loci) while the lowest NP was 63 in the Shahroud population (12.55% polymorphic loci). The observed number of alleles (Na) ranged from 1.342 to 1.125, and the effective number of alleles (Ne) varied from 1.228 in Qazvin to 1.091 in Shahroud population. The mean values for Na and Ne were 1.248 and 1.165, respectively. Likewise, the values for both Shannon's information index and Nei's gene diversity were highest in the Qazvin accessions (0.197 and 0.132, respectively). The Shannon's information index and Nei's gene diversity were lowest in the Shahroud accessions (0.075 and 0.051, respectively).

**Table 2. Estimated genetic diversity of nine walnut germplasm populations.**

| Population | Sample Size | PLP | NP | Na | Ne | H | I |
|---|---|---|---|---|---|---|---|
| Alborz | 21 | 34.07% | 172 | 1.342 | 1.219 | 0.127 | 0.187 |
| Kerman | 13 | 28.88% | 145 | 1.288 | 1.180 | 0.104 | 0.174 |
| Shahroud | 3 | 12.55% | 63 | 1.125 | 1.091 | 0.051 | 0.075 |
| Urmia | 10 | 25.70% | 129 | 1.257 | 1.168 | 0.096 | 0.141 |
| Tabriz | 12 | 26.89% | 135 | 1.268 | 1.182 | 0.102 | 0.150 |
| Touyserkan | 3 | 17.13% | 86 | 1.171 | 1.122 | 0.069 | 0.100 |
| Qazvin | 33 | 37.05% | 186 | 1.370 | 1.228 | 0.132 | 0.197 |
| USA | 5 | 22.91% | 115 | 1.229 | 1.158 | 0.089 | 0.130 |
| Europe | 4 | 18.84% | 91 | 1.181 | 1.136 | 0.075 | 0.109 |
| Mean | | 25.05% | 125 | 1.248 | 1.165 | 0.094 | 0.140 |

**PLP**: Percentage of polymorphic loci, **NP**: Number of Polymorphic loci, **Na**: Observed number of alleles, **Ne**: effective number of alleles (Kimura and Crow (1964), **I**: Shannon's index, **H**: Nei's (1973) gene diversity index.

## Population structure analysis

To determine the structure of walnut populations and the genetic relationship among samples, analyses of- population structure, cluster analysis, and principal coordinate analysis, (PCoA) were performed. Cluster analysis provides an easy and effective way to evaluate genetic diversity [27]. AFLP data using the RB algorithm, grouped the accessions into six clusters (Fig 1). The highest and lowest number of accessions were belong to cluster 1 (40 accessions) and cluster 3 (10 accessions). Internal similarity (ISim) and external similarity (Esim) for K = 6, along with the sample size, are presented in S3 Table. Cluster 4 had the highest value for internal similarities (0.580), while Cluster 3 had the lowest (0.389) amount. Fig 1 shows a mountain visualization of the results for the six clusters.

As indicated by the distances between peaks, Cluster 6 had the lowest value for external similarities and was the farthest group from the other clusters. Cluster 4 had the lowest value for internal similarity.

To characterize collection structure, principal coordinate analysis (PCoA) was performed on the dataset of 104 genotypes/cultivars (S1 Fig). PCoA based on a similarity matrix explained 19.96% and 27.74% of variance on the first and second axis respectively (S1 Fig).

The genetic structure of walnut germplasm was analyzed by STRUCTURE software. Then the STRUCTURE output was submitted to STRUCTURE HARVESTER software to obtain the most likely K value. A clear pinpointed peak at K = 3 was observed, which classified the 104 accessions into three main groups (Fig 3). Fig 3 illustrates the level of admixture of each individual in the population. The first genetic group (in red) contained 21 individuals from Qazvin and Alborz. The second group (in green) consisted of 43 individuals, mainly from Touyserkan, Europe, Tabriz and a few from Qazvin. The third (in blue) included 50 individuals from Kerman, Shahroud, Urmia and some of the genotypes from Alborz.

The separation of populations by origination has not been seen typically in Persian walnut [23, 28]. For classification using multivariate methods, the cluster analysis (Fig 1) displays more complexity than STRUCTURE analysis. In general, both STRUCTURE and cluster

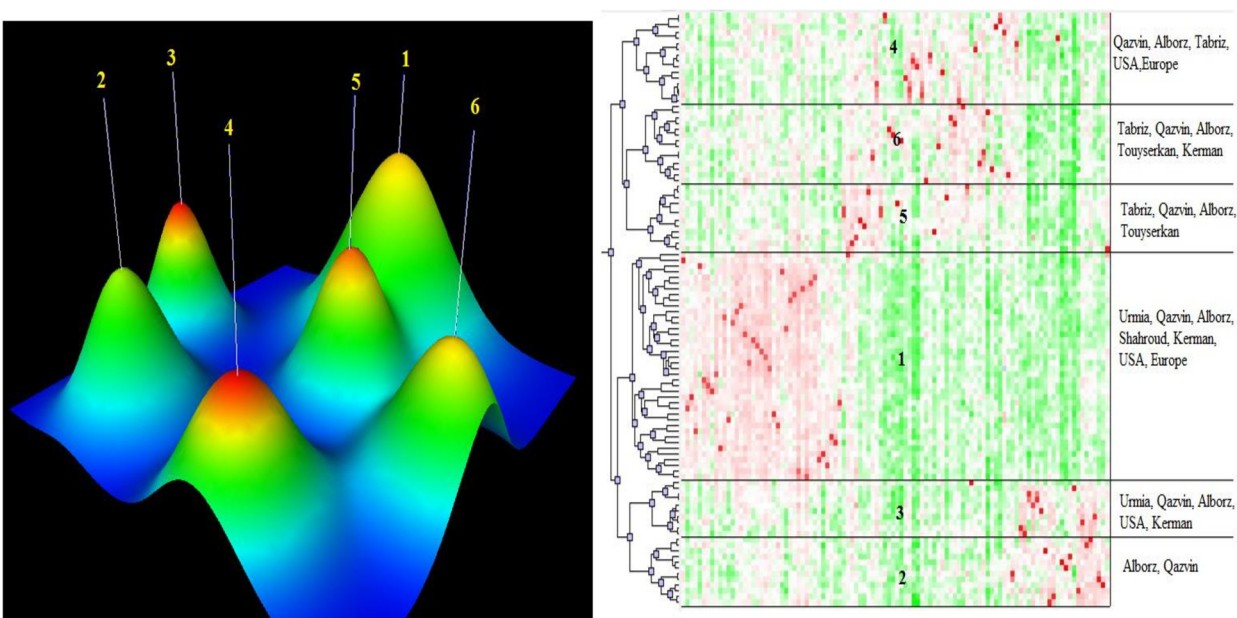

**Fig 1. Mountain visualization of k-means clustering analysis combined with multidimensional scaling.**

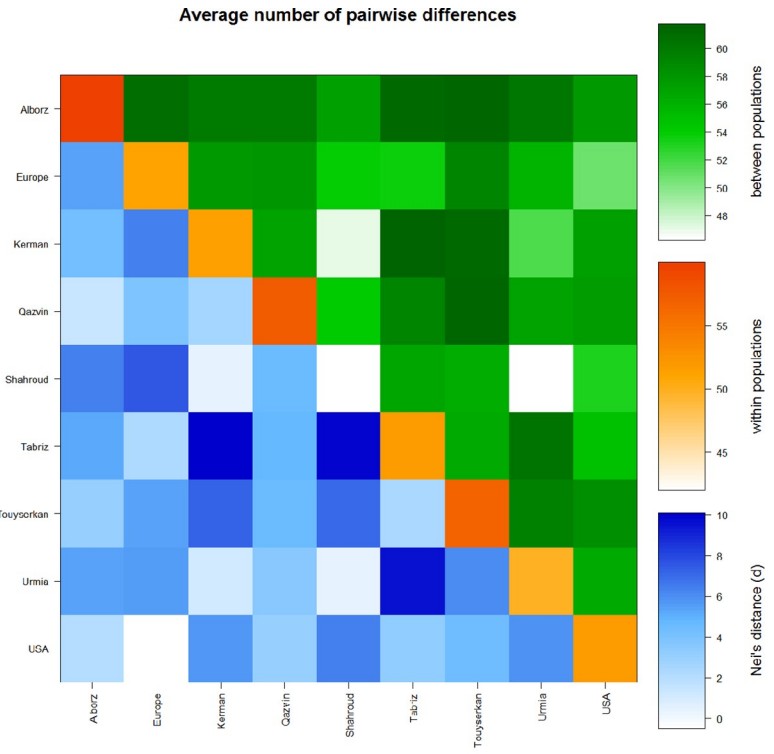

**Fig 2. Nei's genetic distance coefficient for the nine walnut populations.**

analyses showed the same strong genetic division among walnut germplasm (Fig 1). Our results agreed with Ebrahimi et al [13], compared the genetic diversity of *Juglans regia* L. growing in the cold temperate region of the eastern U.S.A. with *J. regia* growing in the cold Mediterranean regions of Europe. Their results indicated that "Early Mature" walnuts were exhibiting relatively high levels of genetic diversity and accessions were genetically different from "Normal Growth" group.

Partitioning the variation within and between populations using the analysis of molecular variance (AMOVA) showed that 93.98% of the genetic variability was existed within while 6.32% of variation was between populations (S4 Table). Similarly, Wang et al [29] found that 81.4% of genetic diversity was within and 18.6% was between the walnut populations from Central and South western China. Nei's genetic distances for these nine populations were shown in Fig 2.

## Qualitative phenotypic traits

S5 Table displays the range, median and mean values for the evaluated nut traits, their coefficient of variation and the Shannon index values. Among the analyzed characters, the shape in longitudinal section through the suture showed the highest coefficient of variation (CV = 57.17%). Light kernel color has high economic value and is therefore very important in selection of new cultivars [25]. The median value for kernel color was 5 (medium color) with CV = 34.76%. Other important kernel traits are kernel size and kernel removal. The median for kernel removal was 3, indicating the kernels of most genotypes easily separable from the shell. Adherence of the two halves of the shell is another important trait [25]. Nuts with poor shell seal are more easily damaged by pests during storage [30]. The median for this trait was 5

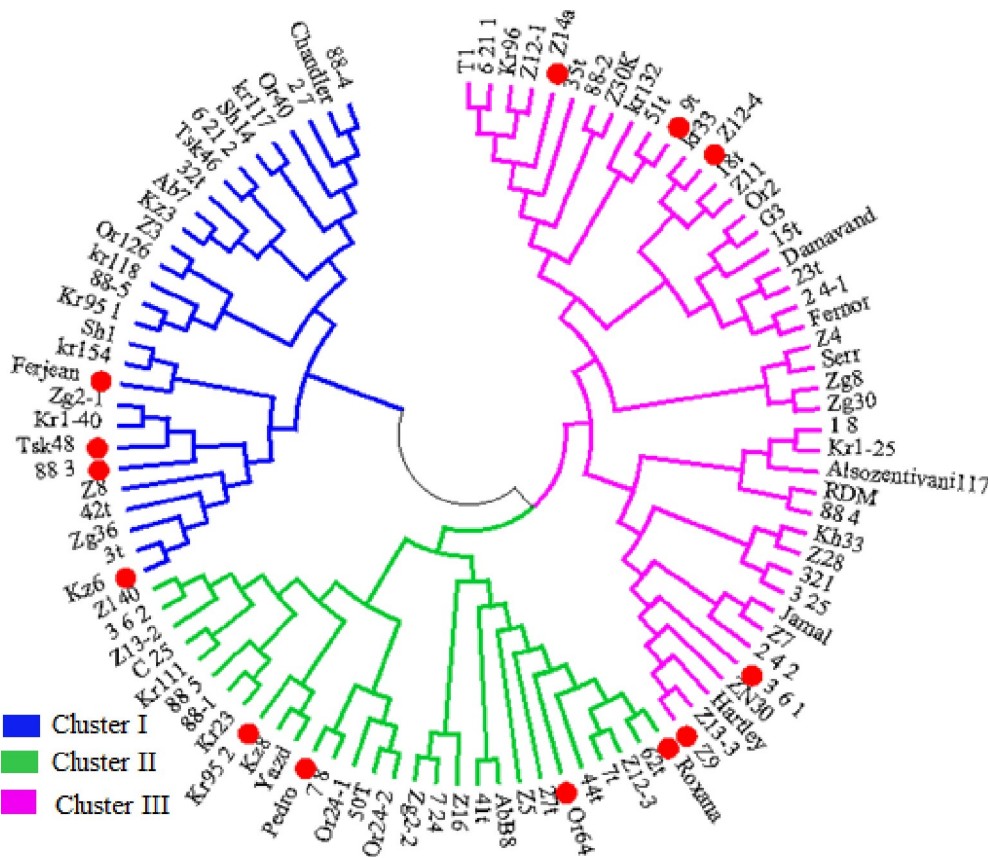

**Fig 3. Genetic classification of *J. regia* accessions using the Neighbor-Joining method.** The genotypes selected for a core collection are illustrated in red.

(medium) with CV = 33.02%. All these traits have been considered in selecting promising genotypes for walnut breeding programs [31].

Among the analyzed traits, the shape in longitudinal section through suture showed the highest Shannon diversity index (H′ = 0.934) (S5 Table).

For explanation of the measured character symbols, see S1 Table

The dendrogram for nut traits, based on neighbor-joining method, classified the accessions into three major clusters (Fig 3).

## Development and evaluation of core collections

To determine a core collection, initial selections were made based on use of AFLP (CC1), quantitative phenotypic traits (CC2), and qualitative phenotypic data (CC3). Then, these three core collections (CC1–CC3) were merged to generate a composite core collection (CC4) (Fig 4). Kumar et al. [5] suggested that in order to capture the maximum range of allelic diversity/ traits in a core set, and to prevent trade-off between two data types when used together, it is better to combine phenotypic and molecular variability. Therefore, three core collections, CC1 (27 accessions), CC2 (13 accessions), and CC3 (18 accessions) were combined to form a non-redundant composite core collection referred to as CC4. CC4 was comprised of 46 accessions from Alborz, Kerman, Qazvin, Shahroud, Tabriz, Touyserkan and Urmia, USA and Europe (except shahroud walnut populations (Table 3). The CC4 showed a 100% coverage value for the different phenotypic and genetic variables under consideration.

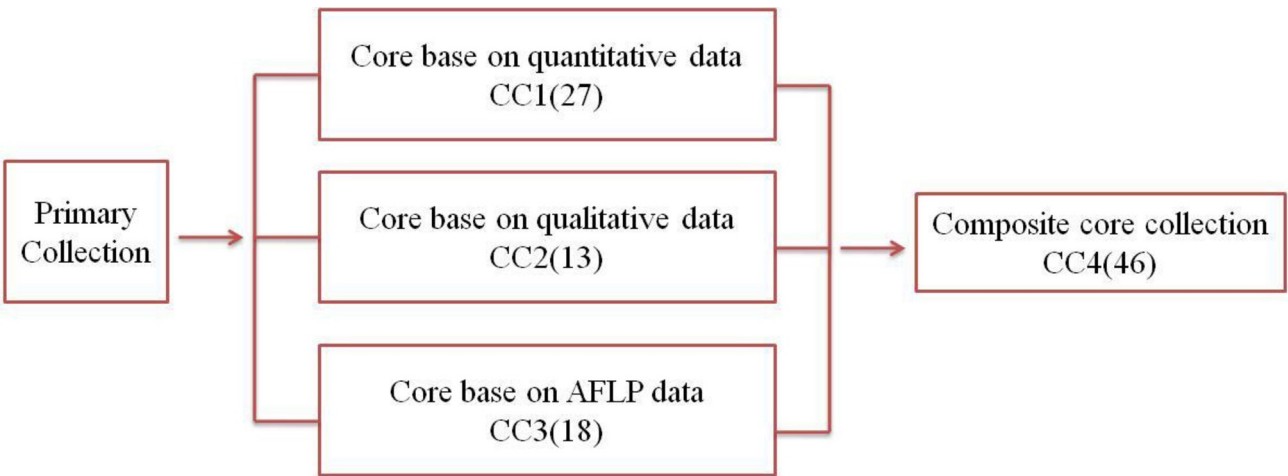

**Fig 4. Flowchart describing the steps for development of a core collection for walnut.** Numerical values indicate the number of accessions in respective cores.

For a core collection to be considered representative, the MD% must be less than 20% and CR% must be > 80% [5]. In addition, more effective core collection must have a lower VD and higher VR (more than 100%) [8]. The composite core collection (CC4) was built using a combination of the genotypic and phenotypic data (Table 3).

The Shannon-Weaver diversity index range (I) for all core collections varied from 0.338 to 0.497. The Nei's genetic diversity (H) ranged from 0.203 to 0.328 (Table 4).

The composite collection, CC4, provided a more logical and exhaustive representation of all the phenotypic and genetic variability of the independent core collections (CC1–CC3).

PCA was performed to validate and confirm the distribution of the four core collections. The distribution of the individuals in these collections was explained by the first two principal components in Fig 5.

## Discussion

Previous studies have shown that AFLP markers could be a tool for characterization of genetic diversity and population structure in walnut [15, 32]. In this study, the phenotypic traits and AFLP molecular markers were combined to characterize the genetic diversity of a walnut collection and to suggest a core collection. The polymorphism detected by the five AFLP primer

**Table 3. Representation from different regions in the developed walnut Core Collections (CC).**

| Regional gene pool | Entire collection | CC1 | CC2 | CC3 | CC4 |
|---|---|---|---|---|---|
| Alborz | 21 | 7 | 1 | 6 | 10 |
| Europe | 4 | 2 | 2 | - | 3 |
| Kerman | 13 | 5 | 1 | - | 6 |
| Qazvin | 33 | 10 | 4 | 7 | 16 |
| Shahroud | 3 | - | - | - | - |
| Tabriz | 12 | - | 1 | 2 | 2 |
| Touyserkan | 3 | - | 1 | - | 1 |
| Urmia | 10 | 1 | 2 | 1 | 3 |
| USA | 5 | 2 | 1 | 2 | 4 |
| Total | 104 | 27 | 13 | 18 | 46 |

**Table 4. Evaluation indices for the developed core collections.**

| Core collection | Variable | Core size | Evaluated parameters | | | | | |
|:---:|:---:|:---:|:---:|:---:|:---:|:---:|:---:|:---:|
| | | | MD% | VD% | VR% | CR% | I | H |
| CC1 | 18 | 27 | 9.88 | 38.37 | 120.28 | 95.05 | 0.338 | 0.203 |
| CC2 | 17 | 13 | 7.50 | 33.26 | 123.92 | 100 | 0.435 | 0.280 |
| CC3 | 197 | 18 | 16.68 | 26.55 | 118.77 | 100 | 0.497 | 0.328 |
| CC4 | — | 46 | 12.03 | 18.11 | 102.64 | 97.74 | 0.411 | 0.262 |

**MD%**: mean difference, **VD%**: variance difference, **VR%**: variable rate, **CR%**: coincidence rate of range, **I**: Shannon's diversity index, and **H**: Nei's genetic diversity.

combinations used in this study is higher than was reported by some researchers [32, 33] while less than another [34]. Nicese et al [35], using 18 RAPD primers and 19 walnut genotypes, observed 23 polymorphic fragments corresponding to about 25% of the polymorphism. Dadras et al [14], using 20 RAPD primers, scored 3.1 polymorphic bands per primer in characterizing of 82 walnut accessions. Pop et al [36] obtained 76.3% polymorphism using 25 RAPD primers in 20 walnut accessions.

The three nucleotide extensions (M-GAG and M-CAG) can also be used to develop Sequence Tagged Site (STS) markers for the identification and tagging of the germplasm. In order to determine the utility of these markers, Polymorphic Information Content (PIC), Resolving Power (RP) and Marker Index (MI) were calculated [37]. The primer combinations used in this study exhibited RP values in the range of 12.13–17.31 (Table 1). The observed range of RP values for the AFLP primer combinations was greater than the result obtained by other researches [32, 38]. Additionally there was a strong linear relationship between the ability of a primer combination to distinguish genotypes and RP values [23]. The primer combination of E-TG × M-CAG, with the highest RP value and polymorphism, was determined to be the most informative combination for estimating the genetic-diversity. This primer combination also had the highest RP value and polymorphism in apricot and peach [39].

In this study the mean higher PIC value than previously observed [32], indicated higher variation among these walnut genotypes. For dominant markers, such as AFLP, estimated marker index in combination with PIC value has been used to assess the informativeness of markers. Because AFLP markers provide a large number of polymorphic fragments, they can assist efficient evaluation of genetic diversity and provide a valuable tool for breeding programs [40].

## Population differentiation and structure

Based on the qualitative phenotypic traits, the 104 walnut genotypes classified into three major clusters using the Neighbor-Joining method, while the mountain visualization of k-means clustering method with AFLP data grouped them into six clusters. Both AFLP and the phenotypic method, with classifying the genotypes of different origins together, showed no clear relationship with geographic origin. Principal coordinate and Structure analysis based on AFLP data also produced three groups.

Structure analysis is a widely used method for inference of hidden population structure in plant species [41]. In this study, three major subpopulations were identified which were not corresponding with geographical origin.

PCoA clustered these genotypes into three main groups and confirmed the K value of the structure analysis.

Structure analysis and PCoA showed similar genetic divisions among the sampled sites that is similar to others reports [13, 42]. The poor association between the molecular marker

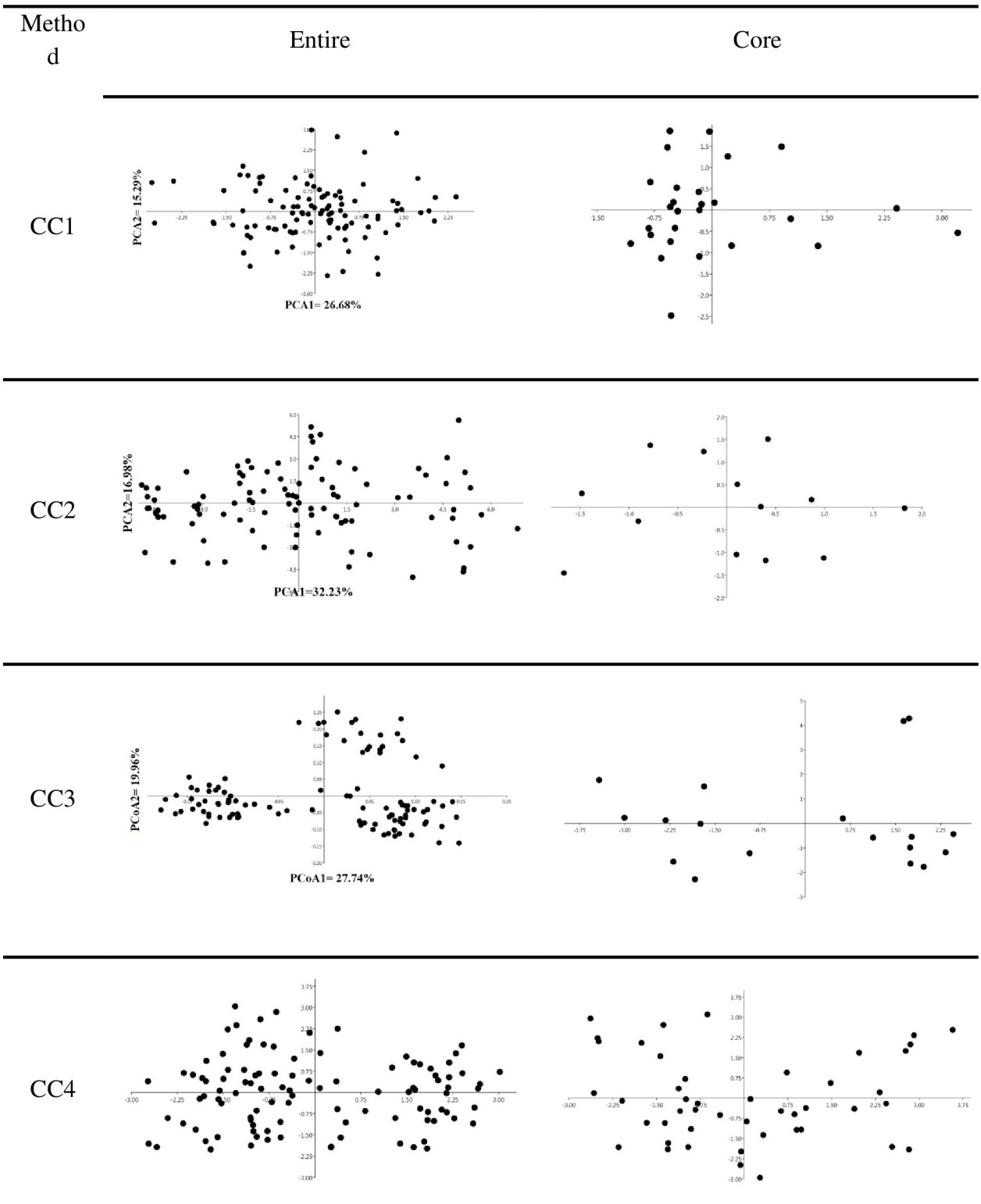

**Fig 5. PCA graphs depicting spread of members of the composite and the three independent core collections.**

data and the geographic origin of the genotypes, has also been reported in previous studies [34, 43].

There were few differences in genetic diversity parameters between the nine studied walnut populations. The AMOVA attributed more than 93.98% of the diversity to individuals, a level similar to that found by Aradhya et al [42] (86%), Ebrahimi et al [13] (85%), and Christopoulos et al [44] (89%).

## Development and size of core collections

In recent years, considerable advances in molecular markers have enabled their utilization for development of core collections [4, 5, 45]. It is clear that, the molecular markers are useful more, when would be used together with the morphological markers. Molecular markers just reflect the DNA attributes, while the morphological markers can be affected by genetic and by epigenetic variation which is not considered if DNA will be used alone.

There are a lot of methods for producing of core collections, including the random method, principal component scoring, the distance-based methods such as Core Hunter, the Maximum Length Sub Tree method (MLST), maximization of the allelic diversity using MSTRAT and PowerCore [46]. Some studies have emphasized use of a maximization strategy for development of robust core collections [47]. Maximization with heuristic searching is considered to be a powerful approach for maintaining diverse and maximum number of alleles at each locus [2]. PowerCore programs have been used successfully to construct core collections with high genetic diversity for various plant species [4, 6, 48].

Various additional information has been used to form core collections, including phenotypic and ecogeographical traits and molecular markers, either alone or in combination [3, 5]. Use of genotypic or phenotypic information alone for the establishment of core collections may not efficiently capture the entire genetic diversity of a species. Therefore, a combination of them was used in our study for the construction of a walnut core collection.

In our study, the four generated core collections efficiently captured the entire range of trait variability. Many accessions were common between different core collections. For instance, 12 accessions were common between CC1, CC2 and CC3. Only 9 accessions were unique to CC1, 7 to CC2, and 18 to CC3. The presence of common accessions between core collections using different types of data indicates an overlap in genetic and phenotypic components of accessions [5]. These constitute a subset of genotypes/cultivars that are extremely diverse at both the molecular and phenotypic level.

Kumar et al [5], argued that in order to capture the maximum range of allelic diversity/traits in a core collection and to prevent trade-off between two data types, it is better to combine phenotypic and molecular variability by merging core collections derived from each type of data separately. For this reason, the core collections were merged to derive a more robust and non-redundant composite core collection (CC4) (Fig 4). The indices (MD%, VD%, VR%, CR %, I, H) for CC4 reflect the composite core collection effectiveness in capturing the diversity of the full walnut collection (Table 4).

## Conclusions

This study demonstrates the usefulness of AFLP markers in characterizing the genetic variation and population structure of the walnut collection and use of this information for creating a core collection. This study is the first attempt in walnut, in which the molecular diversity has been used in conjunction with phenotypic data to develop a core collections. The walnut core collection will provide access to a genetically diverse and important germplasm that can

facilitate characterization of the genetic determinants of trait variability. This information can be used to design more effective breeding programs.

## Supporting information

**S1 Data.**
(XLSX)

**S1 Fig. Principal coordinate analysis of the accessions based on AFLP markers.**
(DOCX)

**S2 Fig. Pattern of individual assignments into three subsets (K = 3) using the STRUCTURE software.** Each individual is shown by a vertical line with one to three colored segments, according to its estimated membership probabilities (Q).
(DOCX)

**S1 Table. Descriptors for the qualitative traits utilized.**
(DOCX)

**S2 Table. Sequences of oligonucleotide adaptors and primers used for AFLP.**
(DOCX)

**S3 Table. Internal and external similarity measures of groups and membership of walnut in each cluster corresponding to Fig 1.**
(DOCX)

**S4 Table. Analysis of molecular variance (AMOVA) on based on AFLP markers of 104 accessions.**
(DOCX)

**S5 Table. Range median, mean, coefficient of variation and Shannon Diversity Index for the traits evaluated.**
(DOCX)

## Acknowledgments

We gratefully acknowledge Dr. R. Ghaffari, Mis. Farsi and Dr. A. Soleimani for their cooperation.

## Author Contributions

**Conceptualization:** Mohammad Reza Dadpour.

**Data curation:** Razieh Mahmoodi, Mohammad Reza Dadpour, Darab Hassani, Mehrshad Zeinalabedini, Elisa Vendramin.

**Formal analysis:** Razieh Mahmoodi, Mohammad Reza Dadpour, Darab Hassani, Mehrshad Zeinalabedini.

**Funding acquisition:** Darab Hassani.

**Investigation:** Razieh Mahmoodi, Darab Hassani.

**Methodology:** Mohammad Reza Dadpour, Darab Hassani.

**Project administration:** Darab Hassani.

**Resources:** Darab Hassani.

**Software:** Razieh Mahmoodi, Darab Hassani.

**Supervision:** Mohammad Reza Dadpour, Darab Hassani.

**Validation:** Darab Hassani, Charles A. Leslie.

**Visualization:** Darab Hassani, Charles A. Leslie.

**Writing – original draft:** Razieh Mahmoodi, Mohammad Reza Dadpour, Darab Hassani, Charles A. Leslie.

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
