## [Decision Letter · Decision Letter 0]

1 Sep 2020

PONE-D-20-22024

Composite core set construction and diversity analysis of Iranian walnut germplasm using molecular markers and phenotypic traits

PLOS ONE

Dear Dr. Darab Hassani,

Thank you for submitting your manuscript to PLOS ONE. After careful consideration, we feel that it has merit but does not fully meet PLOS ONE’s publication criteria as it currently stands. Therefore, we invite you to submit a revised version of the manuscript that addresses the points raised during the review process.

We look forward to receiving your revised manuscript.

Kind regards,

Ji-Zhong Wan

Academic Editor

PLOS ONE

Journal Requirements:

"This work was supported by the Horticultural Sciences Research Institute (HSRI), Agricultural Research, Education and Extension Organization (AREEO), Systems Biology Department, Agricultural Biotechnology Research Institute, University of Tabriz"

Reviewers' comments:

Reviewer's Responses to Questions

**Comments to the Author**

1. Is the manuscript technically sound, and do the data support the conclusions?

Reviewer #1: Yes

Reviewer #2: Partly

2. Has the statistical analysis been performed appropriately and rigorously? 

Reviewer #1: Yes

Reviewer #2: Yes

3. Have the authors made all data underlying the findings in their manuscript fully available?

Reviewer #1: No

Reviewer #2: No

4. Is the manuscript presented in an intelligible fashion and written in standard English?

Reviewer #1: No

Reviewer #2: No

5. Review Comments to the Author

Reviewer #1: The manuscript describes the population structure of 104 Persian walnut accessions by using AFLP markers in combination with 20 phenotypic variability for 17 qualitative traits. They also applied an advanced maximization strategy with a heuristic approach to develop the core collection. The manuscript brings interesting information of walnut genetic diversity and a new approach for core collection development. However, some points should be improved to the manuscript be acceptable for publication. Writing of the manuscript must be improved. The whole paper would benefit from an English editing. The manuscript needs major revision.

Line 32-34: “This core collection will facilitate identification of genetic determinants of trait variability and aid effective utilization of diversity in walnut breeding programs.”

This phrase is confuse. Can a core collection identify genetic determinants of trait variability?

Line 80: Check the number of evaluated traits. 18 or 17?

Line 86: The AFLP protocol must be well described. The amount of DNA used for digestion with restriction enzymes must be mentioned as well as the modifications of the basic protocol (VOS et al.).

Line 92-93: “Primers used are complementary to the adapters and restriction site with three more selective bases during pre-amplification stage.”

This phrase must be clarified. Usually, the three selective bases at the end of the primer is used during selective amplification according to the traditional AFLP protocol. Please check this out.

Line 108: “STRUCTURE software ver. 2.3.4 was used to analyze the population structure of the full 109 germplasm collection.” Please describe the parameters used to run the program (number of burning periods, MCMC , etc…)

Line 135-136: “Five AFLP primer combinations detected a total of 499 fragments, which included 197 polymorphic among the analyzed accessions.” The number of polymorphic markers (informative) is small. Why only 5 AFLP primer combinations were screened?

Line 193: “Arlequin software was also used to estimate the genetic diversity within and between population”. This information must be added to the Material and methods

Line 241-242: “Previous studies have shown that AFLP markers could be a tool for characterization of genetic diversity and population structure in walnut [15]”. I suggest to include more than one reference, as the author mentioned “Previous studies”.

Line 258: “The primer combination of E-TG × M-CAG..” The selective primer for Eco has only two selective nucleotide at the 3´end? If yes, this must be informed in the material and methods.

Line 269-270: “The 104 genotypes grouped into three major clusters, but the AFLP grouped them into six clusters”. Explain, it is not clear.

Line 286: The size of the core collection must be informed.

Reviewer #2: My first question is whether it is really of value to develop a core collection when there are only 104 accessions. In addition, 46 individuals are selected for the core, which is a large proportion of the original set (44%). Normally a core is about 10% of the population. The problem with the approach of combining three core collections, based on (1) AFLP, (2) quantitative and (3) qualitative data is that the final combined core is substantially larger than an ideal core. This is not discussed anywhere in the manuscript. The manuscript does not give details as to what criteria are used to select the different cores, and the percentage selected. I think the approach needs to be fine-tuned to reduce the total number of individuals in CC4 if this is going to be called a proper core collection. This could be done by doing a final selection on CC4, or reducing the percentage selected in CC1-3.

The document needs thorough editing. The English is not acceptable as it stands. Much more detail needs to be given throughout to add clarity to the narrative.

Abstract: Need to mention the final number of accessions in the core collection

Introduction: You need to mention the breeding system of the walnut as whether it is inbreeding or outcrossing really affects population structure.

Ln 20: What about the quantitative traits?

Ln21: NPB, PIC, MI and I Expand these in the abstract

Ln22: MANOVA expand in abstract

Ln39: Outcrossing? Inbreeding?

Ln38: the Iranian Plateau – sentence needs editing

Ln70: Give geographical co-ordinates of main collection

Ln 75: Some foreign cultivars – how many? Be precise. What was the purpose of this?

Ln 80: What were the qualitative traits?

Ln 148: How were the nine populations defined and what were they? This needs to be included in the methodology section.

Ln 182: I don’t think you can generalize like this for other crops. In many crops diversity is associated with geographical origin. This needs to be deleted and restricted to walnut.

Ln 182: I don’t think you are assigning individuals to geographic groups. This needs to eb corrected.

Ln 187: State what Ebrahimi found.

Ln 193 and 4: Say what was revealed by these parameters

Ln 196: This narrative should not be in the results section, but under the discussion where implications should also be discussed.

Ln 223: eight of the nine geographic populations. Add more detail and clarity throughout the manuscript.

Ln 230: I think this is already clear and no need to repeat

Ln 253: utility of AFLP markers

Ln 262 edit

Ln 269: three clusters based on what?

Ln 270: What analysis supports the lack of clustering with geographic origin? How is this visualized?

Ln 273: Structure analysis is useful for revealing proportions of genome associated with different putative ancestral genotypes. Hidden population structure is not clear.

Ln 274: I didn’t see a correlation analysis. Change this word

Ln 279: associations with geographical location should be put together in one paragraph.

Ln 282: What about diversity based on the three phenotypic groups or six AFLP groups?

Ln 288: I think molecular markers are a useful addition to morphological markers. Molecular markers just reflect DNA but morphological markers can also be affected by epigenetic variation which is not considered if DNA is used alone.

Ln 292: Expand on first use MLST

6. PLOS authors have the option to publish the peer review history of their article (what does this mean?). If published, this will include your full peer review and any attached files.

Reviewer #1: No

Reviewer #2: No

---

## [Author Response · Author response to Decision Letter 0]

14 Nov 2020

Reviewer #1: 

The manuscript describes the population structure of 104 Persian walnut accessions by using AFLP markers in combination with 20 phenotypic variability for 17 qualitative traits. They also applied an advanced maximization strategy with a heuristic approach to develop the core collection. The manuscript brings interesting information of walnut genetic diversity and a new approach for core collection development. However, some points should be improved to the manuscript be acceptable for publication. Writing of the manuscript must be improved. The whole paper would benefit from an English editing. The manuscript needs major revision. 

The writing has been edited again.

Line 32-34: “This core collection will facilitate identification of genetic determinants of trait variability and aid effective utilization of diversity in walnut breeding programs.”

This phrase is confuse. Can a core collection identify genetic determinants of trait variability?

The sentence has been edited as: “The construction of core collection could facilitate the work for identification of genetic determinants of trait variability and aid more effective utilization of diversity caused by outcrossing, in walnut breeding programs”

Line 80: Check the number of evaluated traits. 18 or 17?

The accessions has been characterized for 17 qualitative trais. Mranwhile, they had been assessed for 18 quantitative traits too. 

Line 86: The AFLP protocol must be well described. The amount of DNA used for digestion with restriction enzymes must be mentioned as well as the modifications of the basic protocol (VOS et al.).

A paragraph regarding the AFLP protocol has been added to the manuscript as follow: 

“The AFLP was performed using the method of Vos et al. [19] with modifications, using enzyme combination EcoRI/MseI. The AFLP primer combinations (MseI, EcoRI) were labeled with infrared dyes IRD-700 and IRD-800 at the 5´ end, accompanied by two and three selective nucleotides at the 3´end. Briefly, 5 µ of extracted DNA at a concentration of approximately 50 ng/ µ genomic DNA was digested with EcoRI/MseI (1 U) and incubated at 37 °C for 3 h. The fragments were ligated with T4 DNA ligase to EcoRI and MseI adapters at 37 °C for 3 h followed by 4°C overnight. Ligated DNA was diluted 1:5 with water and used for pre-amplification. Primers used are complementary to the adapters and restriction site with three more selective bases during pre-amplification stage. Selective amplification was carried out using diluted DNA from the pre-amplification reaction and different combinations of IR-700 labeled EcoRI primer and IR-800 MseI primer. The adaptor sequences, pre-selective amplification primers, and selective primers, are listed in Table S2. After an initial denaturation step at 94 °C for 3 min, selective amplification was done for 10 cycles of 30 s at 95 °C, 30 s at 63 °C as touchdown with 1 °C lowering for each cycle, 2 min at 72 °C. PCR was continued by a further 25 cycles of 30 s at 94 °C, 30 s at 54 °C and 2 min at 72 °C, and one final cycle of extension at 72 °C for 5 min. The amplified products were run on a 6.5% polyacrylamide gel using DNA analyzer (LI-COR 4300, USA).” 

Line 92-93: “Primers used are complementary to the adapters and restriction site with three more selective bases during pre-amplification stage.” This phrase must be clarified. Usually, the three selective bases at the end of the primer is used during selective amplification according to the traditional AFLP protocol. Please check this out.

Correction has been done as: “The two and three selective bases at the end of the primer is used during selective amplification according to the traditional AFLP protocol.” 

Line 108: “STRUCTURE software ver. 2.3.4 was used to analyze the population structure of the full 109 germplasm collection.” Please describe the parameters used to run the program (number of burning periods, MCMC , etc…)

The following sentence has been added to the manuscript: “The number of clusters was selected after 10 independent runs of a burn-in period of 100,000 iterations and 100,000 MCMC repetitions for each value of K (k=1-10).”

Line 135-136: “Five AFLP primer combinations detected a total of 499 fragments, which included 197 polymorphic among the analyzed accessions.” The number of polymorphic markers (informative) is small. Why only 5 AFLP primer combinations were screened?

The number of primers initially were 13. So the text has been corrected as: “For determining the genetic variability among the walnut genotypes, 13 AFLP primers were evaluated. Among the evaluated primers, five primer combinations showed polymorphism with a total of 499 total and 197 polymorphic fragments.”

Line 193: “Arlequin software was also used to estimate the genetic diversity within and between population”. This information must be added to the Material and methods

The sentence has been moved and improved in the Material and Methods. 

Line 241-242: “Previous studies have shown that AFLP markers could be a tool for characterization of genetic diversity and population structure in walnut [15]”. I suggest to include more than one reference, as the author mentioned “Previous studies”.

Another reference has been added. 

Line 258: “The primer combination of E-TG × M-CAG..” The selective primer for Eco has only two selective nucleotide at the 3´end? If yes, this must be informed in the material and methods.

The selective primer had 2 and 3 nucleotide at the 3´end, and the related correction has been done in Material and Methods. 

Line 269-270: “The 104 genotypes grouped into three major clusters, but the AFLP grouped them into six clusters”. Explain, it is not clear. 

The sentence has been corrected as: “Based on the qualitative phenotypic characteristics, the 104 walnut genotypes classified into three major clusters, while based on the the AFLP data they has been grouped them into six clusters.”

Line 286: The size of the core collection must be informed.

The subtitle has changed to: “Development and Size of Core Collections.”

Reviewer #2: 

My first question is whether it is really of value to develop a core collection when there are only 104 accessions. In addition, 46 individuals are selected for the core, which is a large proportion of the original set (44%). Normally a core is about 10% of the population. The problem with the approach of combining three core collections, based on (1) AFLP, (2) quantitative and (3) qualitative data is that the final combined core is substantially larger than an ideal core. This is not discussed anywhere in the manuscript. The manuscript does not give details as to what criteria are used to select the different cores, and the percentage selected. I think the approach needs to be fine-tuned to reduce the total number of individuals in CC4 if this is going to be called a proper core collection. This could be done by doing a final selection on CC4, or reducing the percentage selected in CC1-3.

For the establishment of the collection, in the first step, pre-selections were done among more than10000 walnut genotypes from the main walnut producing areas of Iran including Karaj, Qazvin, Tabriz, Urmia, Kerman, Tuyserkan, and Shahroud. So the selection has been done in two steps and the primary accession number was significantly more than 104. 

The document needs thorough editing. The English is not acceptable as it stands. Much more detail needs to be given throughout to add clarity to the narrative.

The revision of the whole text has been done again. 

Abstract: Need to mention the final number of accessions in the core collection

The number of final accession, 46, has been added to the Abstract. 

Introduction: You need to mention the breeding system of the walnut as whether it is inbreeding or outcrossing really affects population structure.

Correction has been made with adding the following sentences to the Intoduction : “As it is known, the walnut is a monoicous species. The existence of protandry that usually cause the out crossing, increase the variability and affects the population structure. This phenomenon together with the sexual propagation, created a huge segregated walnut population in Iran.”

Ln 20: What about the quantitative traits?

The name of quantitative traits that their related results has been reported previousely by Mahmoudi et al [11] has been mentioned on the subsection of Measurement of phenotypic data as: “bud break; start, end and duration of pollen shedding and pistillate flowers receptivity; nut length, width, thickness and roundness; nut and kernel weight; kernel percentage; shell and membrane thickness; and number of nuts to scaffold and trunk cross area.”

Ln21: NPB, PIC, MI and I Expand these in the abstract

Instead of the abbreviated terms of NPB, PIC, MI and I: the“Number of Polymorphic Bands”, “Polymorphic Information Content”, “Marker Index”, and “Shannon’s Information Index”, has been replaced in the Abstract. 

Ln22: MANOVA expand in abstract

The MANOVA has been substituted with Multivariate Analysis of Variance in the Abstract. 

Ln39: Outcrossing? Inbreeding?

The following sentence regarding the outcrossing has been added: “As it is known, the walnut is a monoicous species. The existence of protandry that usually cause the outcrossing, increase the variability and affects the population structure. This phenomenon together with the sexual propagation, in Iran, created a huge segregated walnut population.” 

Ln38: the Iranian Plateau – sentence needs editing

The sentence was revised as: “Iranian plateau has been considered as a center of origin and domestication of this species where it still exhibits a great diversity.” 

Ln70: Give geographical coordinates of main collection

The geographical coordinates of the main collection (35.754888° N and 50.952986° E) has been added to the Material and Methods. 

Ln 75: Some foreign cultivars – how many? Be precise. What was the purpose of this?

The exact number of foreign cultivars was nine that was added to the manuscript. They were considered as a part of available germplasm. 

Ln 80: What were the qualitative traits?

The names of qualitative traits including: “nut size, nut shape in longitudinal section through suture, nut shape in longitudinal section perpendicular to suture, nut shape in cross section, nut shape of base perpendicular to suture, shape of apex perpendicular to suture, prominence of apical tip, position of pad on suture, prominence of pad on suture, width of pad on suture, depth of grove along pad on suture, structure of surface of shell, adherence of two halves of shell, thickness of dividing membranes, ease of removal, intensity of ground color and kernel size”, that are reported in Table S1, were added to the Measurement of phenotypic data. 

Ln 148: How were the nine populations defined and what were they? This needs to be included in the methodology section.

The following sentence has been added to the text for more clarification about the nine grops: “The walnut genotypes were belonging to seven autochthonous origin (Alborz, Kerman, Qazvin, Shahroud, Tabriz, Touyserkan and Urmia), and two foreign groups (USA and Europe)”

Ln 182: I don’t think you can generalize like this for other crops. In many crops diversity is associated with geographical origin. This needs to be deleted and restricted to walnut.

The related correction has been done. 

Ln 182: I don’t think you are assigning individuals to geographic groups. This needs to eb corrected.

The related correction has been done

Ln 187: State what Ebrahimi found.

A related part of foundings, “Their results indicated that ‘‘Early Mature’’ walnuts were exhibiting relatively high levels of genetic diversity and accessions were genetically different from ‘‘Normal Growth’’ group.” 

Ln 193 and 4: Say what was revealed by these parameters

What parameters gives the arlequin?

Based on the first reviewer’ comment, the sentence related to the use of Arlequin, together with adding a brief explanation about the output parameters, has been moved and added to Material and Methods-Data Analysis as: “Arlequin software was also used to estimate the genetic diversity within and between populations. The analysis could be performed using intra-population and inter-population methods. In the first option statistical information would be extracted independently from each population, whereas in the second method, samples would be compared to each other.”

Ln 196: This narrative should not be in the results section, but under the discussion where implications should also be discussed.

The sentence has been deleted. 

Ln 223: eight of the nine geographic populations. Add more detail and clarity throughout the manuscript.

The name of populations: Alborz, Kerman, Qazvin, Shahroud, Tabriz, Touyserkan, Urmia, USA and Europe (except shahroud) has been added. 

Ln 230: I think this is already clear and no need to repeat

The sentence has been eliminated. 

Ln 253: utility of AFLP markers

The word “AFLP” has been added to the sentence. 

Ln 262 edit

The sentence has been edited as: “In this study the mean higher PIC value than previousely observed [31], indicated higher variation among these walnut genotypes.”

Ln 269: three clusters based on what?

The sentence has been reviseed as: “Based on the qualitative phenotypic characteristics, the 104 walnut genotypes classified into three major clusters using the Neighbor-Joining method, while the mountain visualization of k-means clustering method with AFLP data grouped them into six clusters.”

 Ln 270: What analysis supports the lack of clustering with geographic origin? How is this visualized?

Classification of the genotypes from different origins in the same class showed this outcome. So, the sentence has been revised as: “Both AFLP and the phenotypic method, with classifying the genotypes from different origins together, showed no clear relationship with geographic origin.”

Ln 273: Structure analysis is useful for revealing proportions of genome associated with different putative ancestral genotypes. Hidden population structure is not clear. 

The genetic structure of walnut germplasm was analyzed by STRUCTURE software. The STRUCTURE output was submitted to STRUCTURE HARVESTER software to obtain the most likely K value. A clear pinpointed peak at K =3 was observed, which classified the 104 accessions into three main groups (FigS2). 

Ln 274: I didn’t see a correlation analysis. Change this word

Correction has been made using the word corresponding instead of correlated: “In this study, three major subpopulations were identified which were not corresponding with geographical origin.”

Ln 279: associations with geographical location should be put together in one paragraph

Correction has been done. 

. Ln 282: What about diversity based on the three phenotypic groups or six AFLP groups?

The MANOVA that shows the between and withing group variances has been done only for classification of nine group.

Ln 288: I think molecular markers are a useful addition to morphological markers. Molecular markers just reflect DNA but morphological markers can also be affected by epigenetic variation which is not considered if DNA is used alone.

The sentences, has been revised as: “It is clear that, the molecular markers are more useful when will be used together with the morphological markers. Molecular markers just reflect the DNA attributes, while the morphological markers can be affected by genetic and also epigenetic variation which is not considered if DNA will be used alone.”

Ln 292: Expand on first use MLST

MLST was expanded to the Maximum Length Sub-Tree method (MLST).

---

## [Decision Letter · Decision Letter 1]

7 Jan 2021

PONE-D-20-22024R1

Composite core set construction and diversity analysis of Iranian walnut germplasm using molecular markers and phenotypic traits

PLOS ONE

Dear Dr. Hassani,

Thank you for submitting your manuscript to PLOS ONE. After careful consideration, we feel that it has merit but does not fully meet PLOS ONE’s publication criteria as it currently stands. Therefore, we invite you to submit a revised version of the manuscript that addresses the points raised during the review process.

We look forward to receiving your revised manuscript.

Kind regards,

Ji-Zhong Wan

Academic Editor

PLOS ONE

Additional Editor Comments (if provided):

Please revise the manuscript according to the reviewer' s comment.

Reviewers' comments:

Reviewer's Responses to Questions

**Comments to the Author**

1. If the authors have adequately addressed your comments raised in a previous round of review and you feel that this manuscript is now acceptable for publication, you may indicate that here to bypass the “Comments to the Author” section, enter your conflict of interest statement in the “Confidential to Editor” section, and submit your "Accept" recommendation.

Reviewer #1: (No Response)

2. Is the manuscript technically sound, and do the data support the conclusions?

Reviewer #1: Yes

3. Has the statistical analysis been performed appropriately and rigorously? 

Reviewer #1: Yes

4. Have the authors made all data underlying the findings in their manuscript fully available?

Reviewer #1: Yes

5. Is the manuscript presented in an intelligible fashion and written in standard English?

Reviewer #1: No

6. Review Comments to the Author

Reviewer #1: Although the authors have included the reviewer comments in their manuscript, some minor modifications are suggested below:

- I suggest that the authors carefully check the manuscript for typographical or grammatical errors and also all the references. The manuscript has some typographical errors. Some sentences still need to be revised to provide a better understanding of the text.

Abstract:

line 19-20: Not only qualitative traits was used. Thus, I suggest to mention also the 18 quantitative traits. " the phenotypic variability of 17 and 18 qualitative and quantitative traits, respectively. See information on line 246-247 ("To determine a core collection, initial selections were made based on use of AFLP (CC1), quantitative phenotypic traits (CC2), and qualitative phenotypic data (CC3)."

-Please, use the same term throughout the manuscript, i.e. traits. Some sentences use qualitative traits while in others use qualitative characteristics.

Line 99: According to table S2, 9 selective AFLP primer combinations were used. (M-CAT; M-GAG; M-CAG) x (E-CT; E-GT; E-AT) which gives 3X3= 9. Why 13 are mentioned in the manuscript?

Line 104-105: The sentence "The AFLP primer combinations (MseI, EcoRI) were labeled with infrared dyes IRD-700 and

IRD-800 at the 5´ end, accompanied by two and three selective nucleotides at the 3´end." should be corrected according to the Table S2, in which MseI has three selective nucleotides while EcoRI two. Thus the sentence should be rewriten as:...accompanied by three and two selective nucleotides at the 3´end, respectively".

Line 106: The unit microliters should be corrected: "5 µ of extracted DNA at a concentration of approximately 50 ng/ µ genomic".

Line 107-108: "The fragments were ligated with T4 DNA ligase to EcoRI and MseI adapters at 37 °C for 3 h followed by 4°C overnight. ". Check the information regarding the overnight incubation temperature. Is it 4 or 16?

Line 109-110: "Primers used are complementary to the adapters and restriction site with three more selective bases during pre-amplification stage". Please check this statement. The information does not match with the sequences of oligonucleotide adaptors and primers used for AFLP (Table S2).

Line 112: "... different combinations of IR-700 labeled EcoRI primer and IR-800 MseI primer."

Of the two primers combination (EcoRI/MseI), only one labeled primer is used in the reaction, the other one is not labeled. Thus, the sentence should be rewriten to avoid misinterpretation. I suggest to include on Table 1, for each selective primer combination, which one was labeled, i.e., E-CT (IR700)/M-CAT.

Line 145: correct the name POWERCORE

Line 166: Check the initial number of selective primer combinations. (13 or 9?)

Line 225: Between or among populations? Is the 6.32% value an average among the populations?

7. PLOS authors have the option to publish the peer review history of their article (what does this mean?). If published, this will include your full peer review and any attached files.

Reviewer #1: No

---

## [Author Response · Author response to Decision Letter 1]

20 Jan 2021

Dear Ji-Zhong

Thank you and the reviewer for the valuable comments. We made the proposed corrections, except in some cases that there were some clarifications could be found in “Response to reviewer”:

Thank you very much again and best wishes,

Darab Hassani

Corresponding author

---

## [Decision Letter · Decision Letter 2]

3 Mar 2021

Composite core set construction and diversity analysis of Iranian walnut germplasm using molecular markers and phenotypic traits

PONE-D-20-22024R2

Dear Dr. Hassani,

We’re pleased to inform you that your manuscript has been judged scientifically suitable for publication and will be formally accepted for publication once it meets all outstanding technical requirements.

Kind regards,

David D Fang, Ph.D.

Academic Editor

PLOS ONE

Additional Editor Comments (optional):

Reviewers' comments:

Reviewer's Responses to Questions

**Comments to the Author**

1. If the authors have adequately addressed your comments raised in a previous round of review and you feel that this manuscript is now acceptable for publication, you may indicate that here to bypass the “Comments to the Author” section, enter your conflict of interest statement in the “Confidential to Editor” section, and submit your "Accept" recommendation.

Reviewer #1: All comments have been addressed

2. Is the manuscript technically sound, and do the data support the conclusions?

Reviewer #1: (No Response)

3. Has the statistical analysis been performed appropriately and rigorously? 

Reviewer #1: (No Response)

4. Have the authors made all data underlying the findings in their manuscript fully available?

Reviewer #1: (No Response)

5. Is the manuscript presented in an intelligible fashion and written in standard English?

Reviewer #1: (No Response)

6. Review Comments to the Author

Reviewer #1: (No Response)

7. PLOS authors have the option to publish the peer review history of their article (what does this mean?). If published, this will include your full peer review and any attached files.

Reviewer #1: No

---

## [Editor Report · Acceptance letter]

5 Mar 2021

PONE-D-20-22024R2 

Composite core set construction and diversity analysis of Iranian walnut germplasm using molecular markers and phenotypic traits 

Dear Dr. Hassani:

I'm pleased to inform you that your manuscript has been deemed suitable for publication in PLOS ONE. Congratulations! Your manuscript is now with our production department. 

Kind regards, 

on behalf of

Dr. David D Fang 

Academic Editor

PLOS ONE